# What Can We Learn from Unlearnable Datasets?

**Pedro Sandoval-Segura**[1]   **Vasu Singla**[1]   **Jonas Geiping**[1]
**Micah Goldblum**[2]   **Tom Goldstein**[1]
[1]University of Maryland   [2]New York University
{psando, vsingla, jgeiping, tomg}@umd.edu   goldblum@nyu.edu

## Abstract

In an era of widespread web scraping, unlearnable dataset methods have the potential to protect data privacy by preventing deep neural networks from generalizing. But in addition to a number of practical limitations that make their use unlikely, we make a number of findings that call into question their ability to safeguard data. First, it is widely believed that neural networks trained on unlearnable datasets only learn shortcuts, simpler rules that are not useful for generalization. In contrast, we find that networks actually can learn useful features that can be reweighed for high test performance, suggesting that image protection is not assured. Unlearnable datasets are also believed to induce learning shortcuts through linear separability of added perturbations. We provide a counterexample, demonstrating that linear separability of perturbations is not a necessary condition. To emphasize why linearly separable perturbations should not be relied upon, we propose an orthogonal projection attack which allows learning from unlearnable datasets published in ICML 2021 and ICLR 2023. Our proposed attack is significantly less complex than recently proposed techniques.[1]

## 1   Introduction

Deep learning is fueled by an abundance of data, the collection of which is largely unregulated [17, 20, 14, 32]. Images of human faces [9], artwork [1], and text [22, 24] are increasingly scraped at scale without consent. In an attempt to prevent the unauthorized use of data, unlearnable dataset methods make small perturbations to data so that deep neural networks (DNNs) trained on the modified data result in poor test accuracy [10, 7, 6, 36, 27]. The idea is that if generalization performance is harmed by incorporating the modified, "unlearnable" data, third parties will be disincentivized from scraping it. Unlearnable dataset methods answer the following question: *How should one imperceptibly modify the clean training set to cause the largest generalization gap?*

In this paper, we analyze properties of unlearnable dataset methods in order to assess their future viability and security promises. Currently, unlearnable dataset methods are unlikely to be used for safeguarding public data because they require perturbing a large portion (*e.g.* more than $50\%$) of the training set [10, 27] and adversarial training is a principled attack [31]. Additionally, published data is immutable and must withstand current and future attacks [23]. While there are a number of reasons why these unlearnable dataset methods are unlikely to be employed, our results shed light on privacy implications and challenge common hypotheses about how they work. We make several findings by analyzing a number of unlearnable datasets developed from diverse objectives and theory. In particular,

- We demonstrate that, in many cases, neural networks can learn generalizable features from unlearnable datasets. Our results imply that while resulting test accuracy may be low, DNNs may still learn useful features from protected data.

---

[1]Code is available at `https://github.com/psandovalsegura/learn-from-unlearnable`.

37th Conference on Neural Information Processing Systems (NeurIPS 2023).

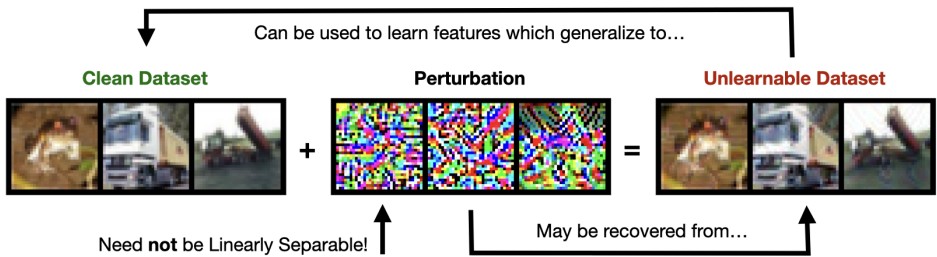

Figure 1: We study unlearnable datasets constructed from additive perturbations. Although unlearnable datasets are created to prevent the exploitation of the clean dataset, we show that unlearnable datasets can be used to learn features which generalize to clean test data. We demonstrate that unlearnable datasets can be created without linearly separable perturbations. And we develop a novel method for recovering class-wise, linearly separable perturbations from unlearnable datasets.

- We challenge the common belief that unlearnable datasets work due to linearly separable perturbations. We construct a counterexample, suggesting that there is room for new unlearnable dataset methods that evade common attacks.
- Inspired by our results on linear separability of perturbations, we present a new Orthogonal Projection attack which allows learning from unlearnable datasets perturbed by class-wise, linearly separable perturbations. Our attack demonstrates that class-wise applied perturbations do not protect data from being learned. Our attack is often more effective than adversarial training, at a fraction of the computational cost.

**Terminology**   Data poisoning methods that perturb the entire training dataset, and which we refer to as "unlearnable datasets" or simply "poisons", are also known as availability attacks [7, 36], generalization attacks [37], delusive attacks [31], or simply unlearnable examples [35]. Throughout this work, unlearnable datasets are considered defenses, given that their primary use case is to prevent the exploitation of data. Aiming to learn from unlearnable datasets is considered an attack.

## 2   Related Work

**Unlearnable Datasets**   One of the earliest instances of a data poisoning attack intended to reduce overall test performance is from [2], who optimize a single sample that corrupts SVM training. Poisoning a convex model like SVM or logistic regression can be solved exactly, but poisoning deep neural networks is more challenging. A number of approaches have been proposed, validated primarily by empirical evidence due to the complexity of the objective (See Eq. 1 and 2). [5] propose using an auto-encoder-like network to generate perturbations against a victim classifier, [37] use NTKs to understand network predictions and optimize perturbations to produce misclassifications, [36] generate linearly separable perturbations, [10, 28] optimize error-minimizing noise whereas [7] optimize error-maximizing noise, and [27] generate autoregressive patterns, among other methods. The diversity of approaches motivates us to understand whether these unlearnable datasets share any properties in common, as we explore in Section 4.3.

**Privacy Vulnerabilities**   In the context of image classification, unlearnable datasets are said to protect privacy by ensuring the modified data is not used or included as part of a larger dataset. That is, training on only poisoned data should prevent test set generalization and training on a dataset consisting of clean and poisoned data should be no better than training just on the clean portion. If poisoned training data is rendered useless for test set generalization, the data will not be used. In this way, only the initial owner can train on the original, unperturbed data and achieve high generalization performance. In recent work, [19] introduce the ability to "lock" and "unlock" training data by leveraging a class-wise perturbation, removable only by someone with knowledge of the exact perturbation. Unfortunately, as we empirically demonstrate in Section 4.4, class-wise perturbations can give a false sense of security. An argument against the use of poisoning for data privacy is the unavoidable reality that published data is immutable, and thus must withstand current *and* future methods which may exploit the data [23]. While it remains possible to recover relatively high test

accuracy from unlearnable datasets, our experiments also suggest that there is not yet a way to restore test accuracy to levels seen during clean training. Unlearnable datasets are known to be vulnerable to adversarial training [10, 31]. By formalizing unlearnable datasets as finding the worst-case training data within a $\infty$-Wasserstein ball, [31] find that adversarial training [16] is a principled defense and can recover a high degree of accuracy by preventing models from relying on non-robust features. [28] challenge this assertion and optimize perturbations designed to degrade model performance under adversarial training. In Section 4.4, we show that adversarial training remains a strong defense against unlearnable datasets. In fact, we find that hyperparameter changes to the adversarial training recipe allows learning from *Robust Unlearnable Examples* [28], demonstrating the brittleness of unlearnable datasets.

**How Unlearnable Datasets Work**  There are many explanations for how unlearnable datasets prevent networks from test set generalization: error-minimizing perturbations cause overfitting [10], error-maximizing noise stimulates learning non-robust features [7], convolutional layers are receptive to autoregressive patterns [27], and more. There are a variety of explanations because different methods arose from different optimization objectives and theory. But the leading explanation comes from Yu et al. [36], who find near perfect linear separability of perturbations for all the unlearnable datasets they consider. They explain that unlearnable datasets cause learning shortcuts due to linear separability of perturbations. In Section 4.3, we find a counterexample, demonstrating that while linear separability of perturbations may certainly be a property that helps unlearnable datasets function, the property is not necessary.

## 3  Problem Setting

We consider the problem of creating a clean-label unlearnable dataset in the context of a $K$-way image classification task, following [10]. We denote the clean training and test datasets as $\mathcal{D}_{\text{train}}$ and $\mathcal{D}_{\text{test}}$, respectively.

Suppose there are $n$ samples in a clean training set, i.e. $\mathcal{D}_{\text{train}} = \{(x_i, y_i)\}_{i=1}^n$ where $x_i \in \mathbb{R}^d$ are the inputs and $y_i \in \{1, ..., K\}$ are the labels. We consider a unlearnable dataset, denoted $\widetilde{\mathcal{D}}_{\text{train}} = \{(x_i', y_i)\}_{i=1}^n$ where $x_i' = x_i + \delta_i$ is the perturbed or poisoned version of the example $x_i \in \mathcal{D}_{\text{train}}$ and where $\delta_i \in \Delta \subset \mathbb{R}^d$ is the perturbation. The set of allowable perturbations, $\Delta$, is typically an $\ell_p$-ball with radius $\epsilon$ where $\epsilon$ is small enough that $\delta$ is imperceptible.

Unlearnable datasets are created by applying a perturbation to a clean image in either a *class-wise* or *sample-wise* manner. When a perturbation is applied class-wise, every sample of a given class is perturbed in the same way. That is, $x_i' = x_i + \delta_{y_i}$ and $\delta_{y_i} \in \Delta_C = \{\delta_1, ..., \delta_K\}$. When samples are perturbed in a sample-wise manner, every sample has a unique perturbation.

All unlearnable dataset methods aim to solve the following bi-level maximization:

$$\max_{\delta \in \Delta} \mathbb{E}_{(x,y) \sim \mathcal{D}_{\text{test}}} \left[ \mathcal{L}(f(x), y; \theta(\delta)) \right] \tag{1}$$

$$\theta(\delta) = \arg\min_{\theta} \mathbb{E}_{(x_i, y_i) \sim \mathcal{D}_{\text{train}}} \left[ \mathcal{L}(f(x_i + \delta_i), y_i; \theta) \right] \tag{2}$$

Eq. 2 describes the process of training a model on unlearnable data, where $\theta$ denotes the model parameters. Eq. 1 states that the unlearnable data should be chosen so that the trained network has high test loss, and thus fails to generalize to the test set.

## 4  Experiments

### 4.1  Datasets and Training Settings

For all of our experiments, we make use of open-source unlearnable datasets: From *Unlearnable Examples* [10], we use their sample-wise and class-wise poisons. From *Adversarial Poisoning* [7], we use their targeted PGD attack poison. From *Autoregressive Poisoning (AR)* [27], *Neural Tangent Generalization Attacks (NTGA)* [37], *Robust Unlearnable Examples* [28], and *Linearly Separable*

Table 1: **Generalizable features can be learned from unlearnable datasets.** While published unlearnable datasets cause DNNs to train to low test accuracy, the features learned by DNNs can be reweighted using DFR to high test accuracy in many cases. For each unlearnable dataset, we report test accuracy and test loss for the best performing checkpoint after DFR. In gray, we indicate test accuracy improvement/deterioration over DFR on a randomly initialized RN-18. DFR uses $5,000$ clean samples for finetuning. For the majority of unlearnable datasets, peak performance seems to occur early in training. High DFR Test Accuracy, and Low DFR Test Loss, indicates that useful features are learned during training.

| TRAINING DATA | MAX DFR TEST ACCURACY | MIN DFR TEST LOSS |
|---|---|---|
| NONE | 35.97 | 2.379 |
| UNLEARNABLE EXAMPLES [10] | 39.56 (+3.59) | 1.798 |
| ADVERSARIAL POISONING [7] | 69.99 (+34.02) | 1.036 |
| AR ($\ell_2$) [27] | 58.73 (+22.76) | 1.531 |
| NTGA [37] | 57.12 (+21.15) | 1.391 |
| ROBUST UNLEARNABLE [28] | 41.02 (+5.05) | 1.790 |
| LSP [36] | 43.31 (+7.34) | 1.675 |
| OPS+EM [35] | 38.98 (+3.01) | 1.869 |
| ∘ OPS [35] | 47.70 (+11.73) | 1.697 |
| ∘ UNLEARNABLE EXAMPLES [10] | 34.09 (-1.88) | 2.037 |
| ∘ REGIONS-4 [26] | 48.09 (+12.12) | 1.706 |
| ∘ RANDOM NOISE | 66.79 (+30.82) | 1.352 |

*Perturbations (LSP)* [36], we use their main poison. For *One-pixel Shortcut* [35], we use their OPS and CIFAR-10-S poisons, but we refer to their CIFAR-10-S poison as *OPS+EM* for ease of attribution. We also include two class-wise *Regions-4* and *Random Noise* poisons which contain randomly sampled perturbations, following [26]. To poison a $K$-class dataset in a class-wise manner, we need $K$ perturbation vectors. For *Regions-4*, each perturbation is independently created by sampling $4$ vectors of size $3$ from a Bernoulli distribution. Each vector is then scaled to lie in the range $[-\frac{8}{255}, \frac{8}{255}]$. Finally, each of the $4$ vectors are repeated along height and width dimensions to create patches of size $16 \times 16$, which are then arranged side by side to achieve a shape of $32 \times 32$. For *Random Noise* perturbations, we sample an i.i.d. vector, one entry per pixel, from a Bernoulli and scale perturbations to fit the imperceptibility constraint. See Appendix A.5 for image samples for all unlearnable datasets we consider.

Results in this section are for CIFAR-10 [13], and additional results for SVHN [18], CIFAR-100 [13], and ImageNet [25] Subset are in Appendix A.2.2 and A.4.2. Unlearnable datasets constructed with class-wise perturbations are prefixed with ∘, otherwise the perturbations were added sample-wise. For the imperceptibility constraint, the *AR* ($\ell_2$) dataset contains perturbations $\delta$ of size $\|\delta\|_2 \leq 1$. *LSP* has perturbations of size $\|\delta\|_2 \leq \epsilon'\sqrt{d}$, where $d$ is the image dimension and $\epsilon' = \frac{6}{255}$. *OPS* and *OPS+EM* contain unbounded perturbations. All other datasets contain perturbations of size $\|\delta\|_\infty \leq \frac{8}{255}$.

We primarily use the ResNet-18 (RN-18) [8] architecture for training on the above datasets. We include results for additional network architectures including VGG-16 [29], GoogleNet [30], and ViT [4] in Appendix A.2.1 and A.4.1. Training hyperparameters can be found in Appendix A.1.

### 4.2  DNNs Can Learn Useful Features From Unlearnable Datasets

Unlearnable datasets are meant to be unlearnable; models trained on unlearnable datasets should not be capable of generalizing to clean test data. Presumably, the modified, unlearnable dataset protects the clean version of the dataset from unauthorized use in this fashion. But by reweighting deep features, we find that DNNs are still able to recover some generalizable features from unlearnable datasets.

**DFR Method**   The technique of training a new linear classification layer on top of a fixed feature extractor is known as Deep Feature Reweighting (DFR) [12] and it is a method for learning from data with spurious correlations, where predictive features in the train distribution are not present in the test

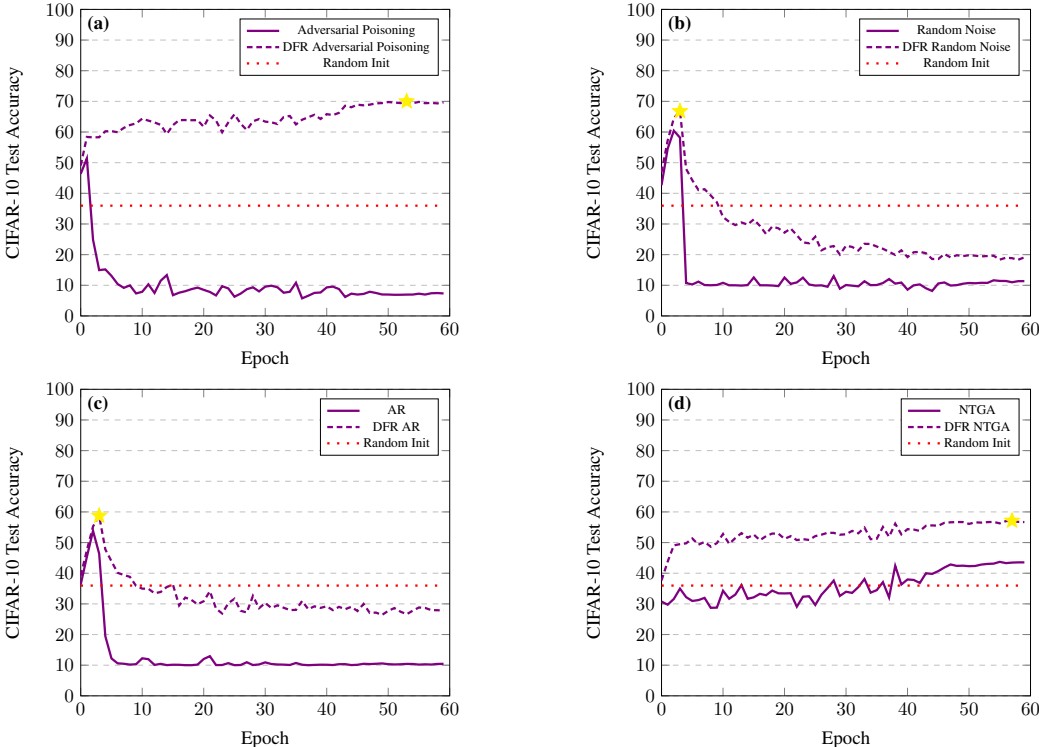

Figure 2: **Features learned during poison training can be reweighed for high test accuracy.**
**(a)** When training on *Adversarial Poisoning*, RN-18 test accuracy (solid line) drops below random chance test accuracy. For the dashed line, each data point at epoch $i$ represents the final test accuracy obtained after DFR using the poisoned checkpoint from epoch $i$ of standard training. Using DFR, the final RN-18 poisoned checkpoint can achieve nearly $70\%$ test accuracy (dashed line). Using DFR on a randomly initialized RN-18 only achieves $40\%$ test accuracy (dotted line). **(b-d)** For different unlearnable datasets, at different points during poison training (marked by star), DFR test accuracy is more than $20\%$ above the accuracy of a randomly initialized feature extractor, suggesting that features learned from unlearnable data are relevant for clean test set generalization.

distribution. We borrow the DFR method to better understand the utility of features learned during poison training. The higher the test accuracy after DFR, the more likely it is that the model has learned features present in the original clean data. If DFR works well on poison-trained networks, this would suggest that even poisoned weights contain a semblance of relevant information for generalization.

To evaluate the extent that generalizable features are learned, we start by saving network weights at every epoch of poison training. In the context of image classification, network weights consist of a feature extractor followed by a fully-connected classification layer mapping feature vectors to class logits. Next, for each checkpoint, we utilize a random subset of $5,000$ clean CIFAR-10 training samples ($10\%$ of the original training set) to train a new classification head (and keep the feature extractor weights fixed). Finally, we plot test accuracy in Figure 2 and evaluate the maximum CIFAR-10 test accuracy achieved through DFR for each dataset and report results in Table 1. As a baseline, we train a classification head on a randomly initialized RN-18 feature extractor and find that these random features can be reweighted to achieve $35.97\%$ test accuracy.

**DFR Results** We find that, to different extents, *DNNs do learn generalizable features* from unlearnable datasets. Surprisingly, on *Adversarial Poisoning* and *NTGA*, RN-18 features actually *improve throughout training*, as we show in Figure 2 **(a)** and **(d)**, despite low test accuracy during poison training (solid lines in Figure 2 **(a)** and **(d)**). On *AR* and ∘ *Random Noise*, test accuracy peaks early in training before dropping to random chance accuracy (solid lines in Figure 2 **(b)** and **(c)**). Still, when early checkpoints are used, one can use features from the poison-trained feature extractors

Table 2: **Linearly separable perturbations are not necessary to create unlearnable datasets.** We train a linear logistic regression model on perturbations and report train accuracy. High train accuracy indicates linear separability of perturbations. Unlike perturbations from other unlearnable datasets, Autoregressive Perturbations (AR) are not linearly separable and are, in fact, less separable than clean CIFAR-10 images.

| Training Data | Train Accuracy |
|---|---|
| Clean | 53.88 |
| Unlearnable Examples [10] | 100 |
| Adversarial Poisoning [7] | 100 |
| AR ($\ell_2$) [27] | **39.58** |
| NTGA [37] | 100 |
| Robust Unlearnable [28] | 100 |
| LSP [36] | 100 |
| OPS+EM [35] | 100 |
| ∘ OPS [35] | 99.93 |
| ∘ Unlearnable Examples [10] | 100 |
| ∘ Regions-4 [26] | 100 |
| ∘ Random Noise | 100 |

to achieve high accuracy (dashed lines in Figure 2 **(b)** and **(c)**). Accuracy curves for *Adversarial Poisoning* are unique because images are perturbed with error-maximizing noise, which correspond to actual features models use during classification [11]. In this case, DFR is reweighting useful, existing features for classification, leading to higher test accuracy. On the other hand, *Random Noise* and *AR* poisons do not perturb images with useful features; instead, both perturb with synthetic noise. In these cases, useful features are still learned during poison training, but only in the first epochs. As training progresses, the model checkpoints are continually corrupted by synthetic noise features which cannot be useful for classification despite reweighting.

As we show in Table 1, ∘ *Unlearnable Examples* are very effective at corrupting network weights during training; using DFR on checkpoints from poison training only yields a maximum test accuracy of 34.09%, nearly 2% worse than a randomly initialized feature extractor. Six of the eleven unlearnable datasets we consider yield checkpoints which achieve DFR test accuracies that are 10% or more higher than a randomly initialized feature extractor. Generally, it is interesting that unlearnable dataset methods with the lowest max DFR test accuracy are those which use error-minimizing perturbations in some capacity, *e.g. Unlearnable Examples*, *Robust Unlearnable Examples*, and *OPS+EM*. Analyzing test loss follows trends from test accuracy: *Adversarial Poisoning*, *AR*, *NTGA*, and *Random Noise* achieve lowest losses – and those poisoned checkpoints also have the highest DFR test accuracy in Table 1. More interestingly, we find that *all* poisoned models have a lower loss than the randomly initialized model.[2] This reinforces our claim that models learned useful features from poisoned data.

Overall, our results suggest that network weights during poison training are not fully corrupted in many cases. While poison-trained networks may be evaluated to have low test accuracy, we show that, for some unlearnable datasets, the networks have learned generalizable features which can be reweighted for high test performance. It is entirely possible that these features, which encode the original data amount to useful image information that the original data owner did not want incorporated into an ML system.

Privacy concerns are often cited as primary motivation for unlearnable datasets, yet our results raise concerns about this framing. A model trained on an apparently unlearnable dataset might have low test error, but this does not imply, as we show, that it does not contain usable information about the original data, and might not keep promises to protect data privacy.

### 4.3 Linearly Separable Perturbations Are Not Necessary

The work of Yu et al. [36] demonstrated that unlearnable datasets contain linearly separable perturbations and suggested that linear separability is a necessary property of unlearnable datasets. However,

---

[2]On 10 class classification task, the expected random chance loss is $-\ln(\frac{1}{10}) \approx 2.302$.

---

**Algorithm 1** Orthogonal Projection

---

**Input:** Unlearnable Dataset, $(X, Y) \in \widetilde{\mathcal{D}}_{\text{train}}$
**Output:** Recovered Dataset, $(X_r, Y)$

 1: **while** not converged **do**
 2:     Sample batch $(x, y)$ from $\widetilde{\mathcal{D}}_{\text{train}}$
 3:     $W \leftarrow W - \eta \nabla_W L(W^T x, y)$
 4: **end while**
 5: Perform QR decomposition on $W$ to obtain $Q$ matrix
 6: $X_r \leftarrow X - QQ^T X$

---

by finding a counterexample, we confirm that linear separability is not necessary to have an effective unlearnable dataset.

Following [36], we train linear logistic regression models on image perturbations by subtracting the clean image from the perturbed image. Given the publication of new unlearnable datasets, we add to Yu et al.'s analysis by including *AR*, *Robust Unlearnable Examples*, and *OPS* poisons. It is possible to obtain image perturbations given our access to clean CIFAR-10, but should not be possible in practice if the clean version of the unlearnable dataset is unavailable to the public. We optimize the logistic regression weights ($3072 \times 10$ parameters for CIFAR-10) using L-BFGS [15] for 500 steps using a learning rate of $0.5$. For reference, we also train a logistic regression model on clean CIFAR-10 image pixels and report the train accuracy in the first row of Table 2. For results on the linear separability of the perturbed *images*, see Appendix A.3.2.

While most current unlearnable datasets contain linearly separable perturbations, as was initially shown by Yu et al. [36], *AR* perturbations stand out as a counterexample in Table 2. After training on *AR* perturbations, a logistic regression model can only achieve $39.58\%$ train accuracy while all other unlearnable datasets we tested contain perturbations that are almost perfectly linearly separable. Because most unlearnable dataset methods developed from diverse optimization objectives result in linearly separable perturbations, we posit that linear separability is an easy solution for optimization of Eq. 1 and 2 to find. For unlearnable datasets that are not optimized, class-wise perturbations are trivially linearly separable, so long as no two class perturbations are the same. Thus, although *AR* perturbations are not linearly separable, they remain separable by a simple 2-layer CNN [27], suggesting that simple perturbations may be a more accurate way of defining effective unlearnable perturbations.

Being the first counterexample of its kind, we believe there may be underexplored unlearnable dataset methods. This finding sheds light on the complexity of unlearnable datasets, whose behavior depends on choices of loss function, optimizer, and even network architecture. An important detail about *AR* perturbations is that they are not optimized – they are generated [27]. We leave to future work an investigation as to why most optimized perturbations in unlearnable datasets to date result in linear separability.

### 4.4 Orthogonal Projection for Learning From Datasets with Class-wise, Linear Perturbations

In this section, we develop a method to learn from unlearnable datasets. Given our understanding of linear separability from Section 4.3, learning the most predictive features in an unlearnable dataset can amount to learning the perturbation itself. We leverage this intuition in our Orthogonal Projection attack, which works by learning a linear model, then projecting the data to be orthogonal to linear weights. In other words, we remove the most predictive dimensions of the data. For unlearnable datsets perturbed by class-wise, linearly separable perturbations, the most predictive dimensions are easily found by a linear model. Attacks for learning from class-wise perturbed unlearnable datasets have included using diffusion models [3], adversarial training [10], and error-maximizing augmentations [21]. In contrast to these techniques, our method is simpler and computationally cheaper.

#### 4.4.1 Orthogonal Projection Method

Unlearnable datasets with class-wise perturbations are so simple that visualizing the average image of a class can expose the class perturbation (see Figure 5). In order to adaptively learn these perturbations and remove them from poisoned data, we propose a method that first learns the simple perturbations and then orthogonally projects samples to omit them. Our Orthogonal Projection method is designed

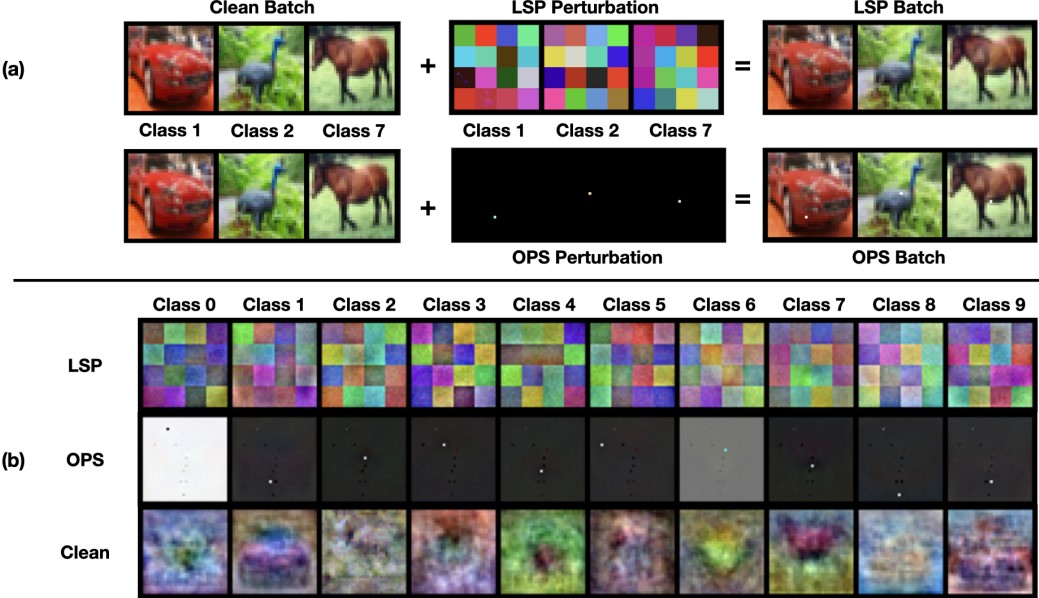

Figure 3: **Our Orthogonal Projection attack learns linearly separable features.** In (**a**), we visualize how LSP [36] and OPS [35] unlearnable data is created by taking clean image and adding a perturbation. In (**b**), we visualize learned weights ($W$ in Algorithm 1) after training a linear model on LSP and OPS images; we also include learned weights after training on clean CIFAR-10 for reference. Learned weights from LSP and OPS unlearnable data nearly match the original perturbation, while learned weights from clean data resemble the corresponding class. In our Orthogonal Projection attack, we project each perturbed image to be orthogonal to each of these learned weights (Algorithm 1, Line 6).

to exploit class-wise perturbations by design and is meant to emphasize why class-wise perturbations should not be relied on for creating unlearnable datasets.

Given an unlearnable dataset, we train a simple logistic regression model on poison image pixels, in an attempt to optimize the most predictive linear image features. The result is a learned feature matrix $W$, on which we perform a QR decomposition such that $W = QR$, where $Q$ consists of orthonormal columns. Image samples $X$ from the unlearnable dataset can be orthogonally projected using the matrix transformation $I - QQ^T$. Orthogonal projection ensures that the dot product of a row of $X$ with every column of $Q$ is zero, so that learned perturbations from the logistic regression model are not seen during the final training run on recovered data. The recovered image samples are then written as $X_r = (I - QQ^T)X$. Finally, we train the final network on recovered data, $X_r$. A detailed algorithm is shown in Algorithm 1. We include additional intuition in Appendix A.4.3.

The added computational cost of Orthogonal Projection over standard training is only in training the logistic regression model and projecting the dataset, which can be done once. For CIFAR-10, we train the logistic regression model for 20 epochs using SGD with an initial learning rate of $0.1$, which decays by a factor of $10$ on epochs $5$ and $8$. We find that training the linear model for longer is often detrimental, potentially because of overfitting.

### 4.4.2 Orthogonal Projection Results

We compare our attack against $\ell_\infty$ adversarial training, as previous work has shown that adversarial training is a principled defense [31] against unlearnable datasets. For adversarial training, we use a 3-step PGD adversary with a perturbation radius of $\frac{8}{255}$ in $\ell_\infty$-norm. Note that large adversarial training perturbation radii harm clean accuracy [33]. We use SGD with momentum of $0.9$ and an initial learning rate of $0.1$, which decays by a factor of $10$ on epoch $75$, $90$, and $100$.

We find our method is *more effective than adversarial training* when learning from unlearnable datasets corrupted in a class-wise manner. In Table 3, our Orthogonal Projection attack leads to greater test accuracy on all class-wise poisons prefixed by ∘. By visualizing the learned weights from

Table 3: **Orthogonal Projection can make class-wise unlearnable data learnable.** Especially for unlearnable datasets with class-wise, linearly separable perturbations, our Orthogonal Projection attack improves CIFAR-10 test accuracy over $\ell_\infty$ Adversarial Training at a fraction of the computational cost. We train RN-18 using different kinds of unlearnable training data and, for every attack, we indicate accuracy improvement/deterioration over standard training (no attack) in gray. For every dataset, we bold the highest test accuracy achieved.

| | | ATTACK | |
| --- | --- | --- | --- |
| TRAINING DATA | NONE | ADV TRAINING | ORTHO PROJ (OURS) |
| CLEAN | 94.37 | 87.16 (-7.21) | 90.16 (-4.21) |
| UNLEARNABLE EXAMPLES [10] | 24.56 | **85.64** (+61.08) | 65.17 (+40.61) |
| ADVERSARIAL POISONING [7] | 7.96 | **85.32** (+77.36) | 14.74 (+6.78) |
| AR ($\ell_2$) [27] | 13.77 | **84.09** (+70.32) | 13.03 (-0.74) |
| NTGA [37] | 40.78 | **84.85** (+44.07) | 82.21 (+41.43) |
| ROBUST UNLEARNABLE [28] | 26.86 | **87.07** (+60.21) | 25.83 (-1.03) |
| LSP [36] | 25.75 | 86.18 (+60.43) | **87.99** (+62.24) |
| OPS+EM [35] | 20.11 | 12.22 (-7.89) | **39.43** (+19.32) |
| ○ OPS [35] | 15.35 | 11.77 (-3.58) | **87.94** (+72.59) |
| ○ UNLEARNABLE EXAMPLES [10] | 14.49 | 85.56 (+71.07) | **89.98** (+75.49) |
| ○ REGIONS-4 [26] | 10.32 | 85.74 (+75.42) | **86.87** (+76.55) |
| ○ RANDOM NOISE | 10.03 | 86.36 (+76.33) | **90.37** (+80.34) |

the linear model in Figure 3, we see that the class-wise, linearly separable perturbations are recovered. Projecting orthogonal to these vectors effectively removes those features from the unlearnable dataset, allowing training to proceed without predictive, but semantically meaningless features. It should not be surprising that Orthogonal Projection is also effective against *OPS* (ICLR 2023), given that all images of a class have the same pixel modified. In Figure 3, we visualize the linear model weights and show that it is able to pinpoint the modified class pixel, whose influence is effectively removed after projection in our method. We visualize additional linear model weights for other datasets in Appendix A.4.5. Orthogonal Projection is also surprisingly effective against *NTGA* (ICML 2021), such that an RN-18 model can achieve $82.21\%$ test accuracy when trained on orthogonally projected *NTGA* images. For seven of the eleven unlearnable datasets we consider, Orthogonal Projection can achieve above $80\%$ test accuracy without adversarial training. Unlike the linear model from Section 4.3 which is trained on only perturbations, the linear model that we train for Orthogonal Projection trains on perturbed images and the resulting data is no longer linearly separable. For this reason, Orthogonal Projection struggles against *Adversarial Poisoning*, *AR*, and *Unlearnable Examples*; the training distribution of the linear model is more complex than just perturbations. Given that *OPS+EM* is a combination of *OPS* and class-wise error-minimizing noise from *Unlearnable Examples*, both Orthogonal Projection and adversarial training are ineffective. For both *OPS* and *OPS+EM*, adversarial training achieves approximately $12\%$ accuracy, a dismal result that is expected given the unbounded *OPS* perturbations. A limitation of our approach is that, by orthogonally projecting data, we effectively remove $K$ dimensions from the data manifold, where $K$ is the number of classes in a dataset. While this may not be a problem for high-resolution images with tens of thousands of dimensions, this detail could impact applicability for low-resolution datasets.

The susceptibility of using class-wise perturbations to craft unlearnable datasets should be expected, given that every image of a class contains the same image feature that is perfectly predictive of the label. Class-wise unlearnable datasets contain simple features that are perfectly correlated with the label and the linear logistic regression model is able to optimize features resembling the original perturbations. By design, our orthogonal projection attack is most effective against class-wise perturbed unlearnable datasets. It serves as evidence that class-wise, linearly separable perturbations cannot be relied upon for protecting data.

## 5   Conclusion

We design experiments to test prevailing hypotheses about unlearnable datasets, and our results have practical implications for their use. First, while all unlearnable datasets cause low test accuracy for

trained models, we demonstrate that generalizable features can still be learned from data. In many cases, high test accuracy can be achieved from a feature extractor trained on unlearnable datasets. If data were modified for protection, an adversary may still be able to train a reasonable feature extractor, a harm for data thought to be secure. We advocate for our evaluation framework to be used for auditing whether new unlearnable datasets prevent useful features from being learned. Second, we find that the reason unlearnable datasets cause low test accuracy is not as simple as previously thought. AR perturbations serve as a counterexample to the hypothesis that unlearnable datasets contain linearly separable perturbations. Our finding that AR perturbations are not linearly separable suggests there could be underexplored methods for protecting data using imperceptible perturbations. Although unlearnable datasets need not have linearly separable perturbations, if they do have them applied in a class-wise manner, we present an Orthogonal Projection attack that can effectively remove them. While learning the perturbations from perturbed images is challenging, our results imply class-wise perturbations *cannot* not be relied upon for data protection. We can learn a significant amount from unlearnable datasets. DNNs can learn generalizable features from unlearnable datasets assumed to be protected, linear separability is not a necessary condition for preventing generalization from unlearnable datasets, and class-wise perturbations can be optimized against and effectively removed.

## Acknowledgments and Disclosure of Funding

This material is based upon work supported by the National Science Foundation under Grant No. IIS-1910132 and Grant No. IIS-2212182, and by DARPA's Guaranteeing AI Robustness Against Deception (GARD) program under #HR00112020007. Tom Goldstein was supported by the ONR MURI program, the AFOSR MURI program, and the National Science Foundation (IIS-2212182 & 2229885). Vasu Singla was supported in part by the National Science Foundation under grant number IIS-2213335. Pedro Sandoval-Segura is supported by a National Defense Science and Engineering Graduate (NDSEG) Fellowship. Any opinions, findings, and conclusions or recommendations expressed in this material are those of the author(s) and do not necessarily reflect the views of the National Science Foundation.

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

# A   Appendix

## A.1   Training Hyperparameters

**DNNs Can Learn Useful Features From Unlearnable Datasets**   In Section 4.2, we train a number of ResNet-18 (RN-18) [8] models on different unlearnable datasets with cross-entropy loss for 60 epochs using a batch size of 128. We save checkpoints at every epoch of training. For our optimizer, we use SGD with momentum of 0.9 and weight decay of $5 \times 10^{-4}$. We use an initial learning rate of 0.1 which decays using a cosine annealing schedule.

For training a new classification layer on feature extractor checkpoints, we use $5,000$ random clean images from the original training set data. Note that $5,000$ images is $10\%$ of CIFAR-10 and CIFAR-100, but $3.9\%$ of Imagenet subset, etc. Following [12], using the feature extractor, we extract embeddings from this clean subset of data and preprocess the embeddings to have mean zero and unit standard deviation. To retrain the last layer, we use the logistic regression implementation from `scikit-learn` (`sklearn.linear_model.LogisticRegression`).

**Linearly Separable Perturbations Are Not Necessary**   In Section 4.3, for every unlearnable dataset, we first gather the set of perturbations by subtracting the clean image from the perturbed image. We zero-one normalize the perturbations before training a linear layer using L-BFGS [15] for 500 steps using a learning rate of 0.5.

**Orthogonal Projection for Learning From Datasets with Class-wise, Linear Perturbations**   In Section 4.4, for CIFAR-10, we train the logistic regression model for 20 epochs using SGD with an initial learning rate of 0.1, which decays by a factor of 10 on epochs 5 and 8 (at epochs which are 0.5 and 0.75 through training). For CIFAR-100, we train the logistic regression model for 60 epochs due to the higher number of classes. After we orthogonally project the unlearnable data using the optimized weights, we train networks using the hyperparameters from checkpoint-training of Section 4.2.

## A.2   Additional Section 4.2 Results: DNNs Can Learn Useful Features From Unlearnable Datasets

### A.2.1   More Model Architectures for Section 4.2

We consider three more model architectures: VGG-16 [29], GoogLeNet [30], and ViT [4]. Our ViT uses a patch size of 4. For RN-18 and VGG-16, feature vectors are 512-dimensional. Feature vectors for GoogleNet are 1024-dimensional. The ViT class token is 384-dimensional.

Table 4: **Generalizable features can be learned from unlearnable datasets, using a variety of network architectures.** We report Max DFR Test Accuracy for each CIFAR-10 unlearnable dataset. In gray, we indicate test accuracy improvement/deterioration over DFR on the corresponding randomly initialized model architecture.

| | MODEL ARCHITECTURE | | |
| CIFAR-10 TRAINING DATA | VGG-16 | GOOGLENET | VIT |
| --- | --- | --- | --- |
| NONE | 35.69 | 48.08 | 37.40 |
| UNLEARNABLE EXAMPLES [10] | 37.84 (+2.15) | 41.08 (-7.00) | 49.57 (+12.17) |
| ADVERSARIAL POISONING [7] | 64.73 (+29.04) | 71.70 (+23.62) | 68.97 (+31.57) |
| AR ($\ell_2$) [27] | 36.98 (+1.29) | 40.12 (-7.96) | 60.53 (+23.13) |
| NTGA [37] | 56.03 (+20.34) | 61.24 (+13.16) | 60.53 (+23.13) |
| ROBUST UNLEARNABLE [28] | 39.13 (+3.44) | 40.59 (-7.49) | 49.10 (+11.70) |
| LSP [36] | 40.86 (+5.17) | 58.22 (+10.14) | 50.95 (+13.55) |
| OPS+EM [35] | 31.31 (-4.38) | 38.57 (-9.51) | 49.73 (+12.33) |
| ○ OPS [35] | 39.63 (+3.94) | 52.02 (+3.94) | 56.04 (+18.64) |
| ○ UNLEARNABLE EXAMPLES [10] | 30.47 (-5.22) | 36.32 (-11.76) | 44.90 (+7.50) |
| ○ REGIONS-4 [26] | 43.29 (+7.60) | 48.65 (+0.57) | 52.60 (+15.20) |
| ○ RANDOM NOISE | 72.08 (+36.39) | 62.19 (+14.11) | 55.58 (+18.18) |

In Table 4, we find that across architectures, *Adversarial Poisoning* data is easiest to extract generalizable features from. Surprisingly, ViT is most effective at learning generalizable features from *all* unlearnable datasets, achieving more than 7% test accuracy improvement over a randomly initialized ViT in all cases. For example, using only 5,000 clean CIFAR-10 samples can be used to achieve nearly 69% test accuracy, while using the same clean samples can only achieve 37.40% test accuracy on a randomly initialized ViT. The GoogleNet architecture weights are seemingly more easily corrupted during training; Max DFR Test Accuracy for *Unlearnable Examples*, *AR*, *Robust Unlearnable*, and other datasets is much lower than test accuracy from a finetuned randomly initialized GoogleNet. Interestingly, the randomly initialized GoogleNet feature extractor achieves the highest DFR test accuracy.

### A.2.2 More Datasets for Section 4.2

We consider three additional base datasets for four unlearnable dataset methods. We use an ImageNet [25] subset of the first 100 classes, following [10]. The train split consists of 129,395 images, while the test split consists of 5,000 images.

Our SVHN [18], CIFAR100 [13], and *Adversarial Poisoning* ImageNet subset datasets contain perturbations of size $\|\delta\|_\infty \leq \frac{8}{255}$. *Unlearnable Examples* ImageNet [25] subset contains perturbations of size $\|\delta\|_\infty \leq \frac{16}{255}$, following their open-source repository. We generate the *Adversarial Poisoning* [7] ImageNet subset from published source code using 1 PGD restart, as opposed to 8 due to computation time. Our SVHN and CIFAR-100 *Adversarial Poisoning* datasets use 3 PGD restarts. On clean SVHN, CIFAR-100, and ImageNet subset, RN-18 achieves 96.33%, 74.14%, 78.92% test accuracy respectively.

Table 5: **Generalizable features can be learned from unlearnable datasets of different underlying distributions.** We report Max DFR Test Accuracy for each unlearnable dataset. RN-18 checkpoints are trained on SVHN, CIFAR-100, and ImageNet subset unlearnable datasets, and DFR is performed using 5,000 clean samples from the corresponding base dataset.

| | FINETUNE DATA | | |
| TRAINING DATA | SVHN | CIFAR-100 | IMAGENET |
|---|---|---|---|
| NONE | 32.05 | 8.13 | 3.64 |
| UNLEARNABLE EXAMPLES [10] | 26.76 | 16.12 | 8.44 |
| ADVERSARIAL POISONING [7] | 87.06 | 44.37 | 20.22 |
| ∘ UNLEARNABLE EXAMPLES [10] | 22.48 | 10.32 | 7.88 |
| ∘ RANDOM NOISE | 27.35 | 47.41 | 23.30 |

In Table 5, we again show that *Adversarial Poisoning* unlearnable data can be easily used to extract generalizable features regardless of the underlying distribution (base dataset of SVHN, CIFAR-100, or ImageNet). For SVHN, *Adversarial Poisoning* is the only dataset from which the trained feature extractor performs better in Max DFR Test Accuracy (87.06%) over a randomly initialized RN-18 (32.05%). As mentioned in Section 4.2, error-minimizing perturbations of *Unlearnable Examples* tend to be most effective at corrupting weights during training, regardless of underlying finetune data.

### A.2.3 Additional Plots for Section 4.2

We add results to the experiment from Figure 2. In Figure 4, unlearnable datasets sufficiently corrupt RN-18 weights during training and prevent DFR from recovering test accuracy.

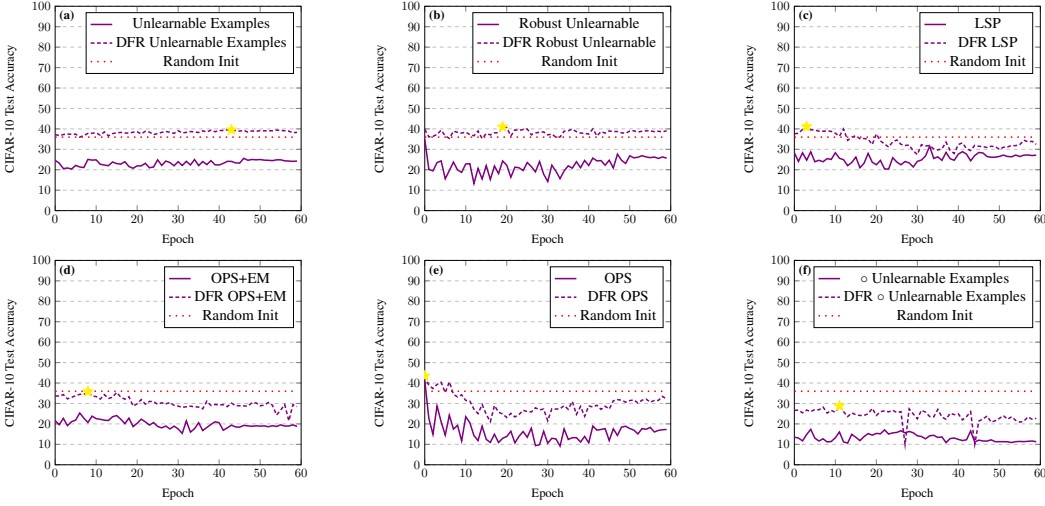

Figure 4: **Representations learned by other poisons we consider are no better than random.** **(a-b)** Reweighting deep features from sample-wise error minimizing noises provide no benefit over random features. DFR on a randomly initialized RN-18 only achieves $40\%$ test accuracy (dotted line). **(c-f)** Other class-wise perturbations are very effective at corrupting network representations during training – so effective that even DFR is unable to recover test accuracy from random features (red dotted line).

## A.3    Additional Results for Section 4.3: Linearly Separable Perturbations Are Not Necessary

### A.3.1    Other Background

A related unlearnable dataset introduces entangled features (EntF) [34] which is motivated by the separability of recent poisoning perturbations. However, separability is qualitatively evaluated through t-SNE visualizations, which is different from the separability experiment we perform. More specifically, t-SNE cluster separability should not be equated to the linear separability we measure in Table 2 because it is possible to have linearly separable data that, when plotted using t-SNE, appears not separable. In other words, the EntF poison from [34] could still contain linearly separable perturbations.

### A.3.2    Evaluating Linear Separability of Poison Images

In Section 4.3, we document the linear separability of perturbations from various poisons, as in [36]. Poison images, on the other hand, behave slightly differently. In Table 6, we report logistic regression train accuracy on various CIFAR-10 poison images. We find that *Unlearnable Examples, LSP, OPS+EM*, and class-wise poisons have linearly separable poison images, but the remaining poisons we consider do not.

## A.4    Additional Section 4.4 Results: Orthogonal Projection for Learning From Datasets with Class-wise, Linear Perturbations

### A.4.1    ViT for Section 4.4

To evaluate recovered data from Orthogonal Projection, we consider an additional architecture: ViT. In Table 7, we train ViT with patch size of $4$ on CIFAR-10 unlearnable datasets using different attacks. Our Orthogonal Projection method is competitive with adversarial training for all class-wise perturbed unlearnable datasets and most sample-wise perturbed datasets. Orthogonal Projection is the best performing method for *OPS+EM* and *OPS* (ICLR 2023). Note that *OPS+EM* and *OPS* are most difficult for adversarial training.

Table 6: **NTGA contains the least linearly separable images, while AR($\ell_2$) images are most comparable to the clean distribution. Other unlearnable datasets become _more_ linearly separable.** We train a linear logistic regression model on poison images and report train accuracy. High train accuracy indicates linear separability of poison images. Mean and one standard deviation computed from 10 independent runs.

| Training Data | Train Accuracy |
|---|---|
| Clean | $53.94 \pm 0.02$ |
| Unlearnable Examples [10] | $100.00 \pm 0.00$ |
| Adversarial Poisoning [7] | $62.40 \pm 0.01$ |
| AR ($\ell_2$) [27] | $53.97 \pm 0.02$ |
| NTGA [37] | $31.48 \pm 0.02$ |
| Robust Unlearnable [28] | $77.21 \pm 0.01$ |
| LSP [36] | $100.00 \pm 0.00$ |
| OPS+EM [35] | $100.00 \pm 0.00$ |
| ○ OPS [35] | $100.00 \pm 0.00$ |
| ○ Unlearnable Examples [10] | $100.00 \pm 0.00$ |
| ○ Regions-4 [26] | $100.00 \pm 0.00$ |
| ○ Random Noise | $100.00 \pm 0.00$ |

Table 7: **Orthogonal Projection can make class-wise unlearnable data learnable for ViT.** Especially for unlearnable datasets with class-wise, linearly separable perturbations, our Orthogonal Projection attack is competitive with $\ell_\infty$ Adversarial Training at a fraction of the computational cost.

| CIFAR-10 Training Data | None | Attack Adv Training | Ortho Proj (Ours) |
|---|---|---|---|
| Clean | 84.99 | 76.38 | 74.14 |
| Unlearnable Examples [10] | 25.39 | 75.44 | 60.15 |
| Adversarial Poisoning [7] | 31.33 | 75.15 | 41.49 |
| AR ($\ell_2$) [27] | 17.13 | 75.12 | 35.11 |
| NTGA [37] | 32.67 | 71.95 | 66.19 |
| Robust Unlearnable [28] | 28.24 | 78.03 | 37.34 |
| LSP [36] | 29.40 | 75.45 | 74.77 |
| OPS+EM [35] | 20.73 | 11.79 | 51.94 |
| ○ OPS [35] | 21.58 | 10.17 | 72.80 |
| ○ Unlearnable Examples [10] | 12.19 | 76.35 | 76.01 |
| ○ Regions-4 [26] | 15.00 | 75.96 | 67.20 |
| ○ Random Noise | 29.66 | 76.23 | 73.05 |

### A.4.2   CIFAR-100 Dataset for Section 4.4

We consider an additional dataset, CIFAR-100, to evaluate Orthogonal Projection. We train a RN-18 on four unlearnable dataset methods. During the first step of Orthogonal Projection, we train the logistic regression model for 60 epochs using SGD with an initial learning rate of $0.1$, which decays by a factor of 10 on epochs 30 and 45.

Table 8: **Orthogonal Projection is competitive on CIFAR-100 class-wise unlearnable data.**

| CIFAR-100 Training Data | None | Attack Adv Training | Ortho Proj (Ours) |
|---|---|---|---|
| Clean | 74.14 | 59.23 | 26.78 |
| Unlearnable Examples [10] | 8.11 | 58.29 | 28.44 |
| Adversarial Poisoning [7] | 5.93 | 57.60 | 25.24 |
| ○ Unlearnable Examples [10] | 1.72 | 60.31 | 41.83 |
| ○ Random Noise | 1.30 | 58.83 | 51.23 |

In Table 8, we find that Orthogonal Projection performs better on class-wise unlearnable data than sample-wise perturbed data, as expected. At approximately the cost of standard training (Attack: None), Orthogonal Projection achives gains of more than $20\%$ test accuracy for sample-wise perturbed data and more than $40\%$ test accuracy for class-wise perturbed data.

### A.4.3   Additional Intuition for Orthogonal Projection

Assume CIFAR-10 images of shape $(3, 32, 32)$. Each column $i$ of $W$ (optimized in Alg 1, Lines 1-4) is a 3072-dimensional vector that represents the most predictive image feature for class $i$. This step serves as recovery of the perturbation. After the QR decomposition of $W$, $Q$ consists of orthonormal columns that form a basis for the column space of $W$. When we say Orthogonal Projection "ensures that the dot product of a row of $X$ with every column of $Q$ is zero," (i.e., $X_r \cdot Q = 0$) this means that every recovered image vector does not contain any linearly separable component (i.e., does not contain any column of $Q$ as a component). Alg. 1, Line 6 ensures image vectors and columns of $Q$ are orthogonal and so the dot product is 0. The "recovered" data thus has 10 dimensions (approximations of the 10 perturbations) removed.

### A.4.4   Subtracting a Class-wise Image

Given that the goal of Orthogonal Projection is to extract perturbations from poison images, it is reasonable to consider visualizing the average image of a class for class-wise poisons like *LSP* and *OPS*. In Figure 5, we see that class-wise average images somewhat reveal class-wise perturbations, but the results are not clear enough to be useful. In contrast, class-wise patters and clearly present in learned weights from logistic regression.

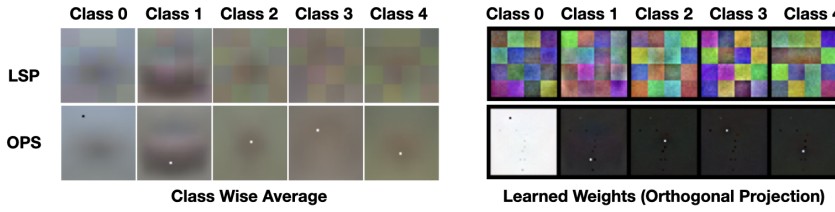

Figure 5: **Average class images display class-wise perturbations, but not as clearly as learned weights from logistic regression.** We compare the average image of a class **(Left)** and learned weights of a logistic regression classifier **(Right)** trained on image pixels (the first step of Orthogonal Projection method) to for *LSP* and *OPS* Poisons. While average image of a class does reveal the class-wise perturbation (block pattern for *LSP* and one highlighted pixel for *OPS*), the result is blurry and contains other semantic image features. Learned weights from Orthogonal Projection properly isolate the perturbation.

For training models on class-wise perturbed data, one might consider subtracting the class-wise average image from each class. However, simply subtracting this average class image from each image does not remove the poisoning effect. Additionally, because we do not know the true class at inference time, we cannot subtract the class image, resulting in a distribution mismatch between train and test sets. This trivial method of subtracting average class images is compared to Orthogonal Projection in Table 9.

### A.4.5   Visualizing Additional Logistic Regression Weights

We visualize additional linear model weights (from the first step of Orthogonal Projection) for sample-wise perturbed unlearnable datasets in Figure 6, and for class-wise perturbed unlearnable datasets in Figure 7. We find that for *Adversarial Poisoning*, *AR*, and *Robust Unlearnable* the linear model learns features comparable to when trained on clean data. We posit that because the diversity of perturbations in these datasets is higher, the linear model struggles to find predictive features to project away. In contrast, for class-wise perturbed data, Figure 7, demonstrates that the linear model can recover features that resemble the original class-wise perturbation.

Table 9: **Subtracting the average class image from training images is not effective.** For *LSP* and *OPS*, datasets with class-wise perturbations, our Orthogonal Projection attack improves CIFAR-10 test accuracy over simply subtracting the average class image at training time.

| | ATTACK | |
|---|---|---|
| TRAINING DATA | CLASS-AVG SUBTRACT | ORTHO PROJ (OURS) |
| LSP [36] | 13.05 | 87.99 |
| ○ OPS [35] | 12.62 | 87.94 |

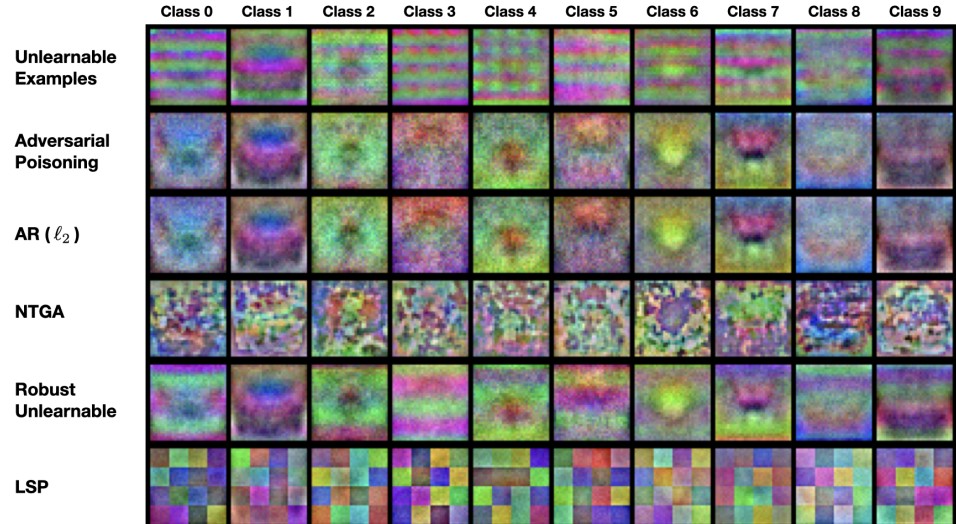

Figure 6: **Learned Weights from first step of Orthogonal Projection on sample-wise perturbed CIFAR-10 unlearnable datasets.** We visualize learned weights ($W$ in Algorithm 1) after training a linear model on unlearnable datasets. Learned weights from *Adversarial Poisoning*, *AR*, and *Robust Unlearnable* resemble the learned weights from clean data (See Figure 3). In our Orthogonal Projection attack, we project each perturbed image to be orthogonal to each of these learned weights (Algorithm 1, Line 6).

## A.5 Samples from Unlearnable Datasets

We visualize samples from sample-wise perturbed CIFAR-10 unlearnable datasets in Figure 8, and from class-wise perturbed unlearnable datasets (prefixed by ○ throughout results) in Figure 9. NTGA is omitted due to data ordering of the publicly available poison. We also visualize SVHN samples in Figure 10 and CIFAR-100 in Figure 11.

## A.6 Broader Impact Statement

Our findings test prevailing hypothesis about unlearnable datasets and our results have practical implications for their use. Two of our three main conclusions relate to privacy vulnerabilities when employing unlearnable datasets for data protection. In one experiment, we demonstrate useful features can be learned from unlearnable data. In another, we demonstrate how one can effectively remove a class-wise perturbation. Our findings highlight the need for extra caution when it comes to using unlearnable datasets. By making this information available to the public, the capabilities and vulnerabilities of unlearnable datasets can be better understood.

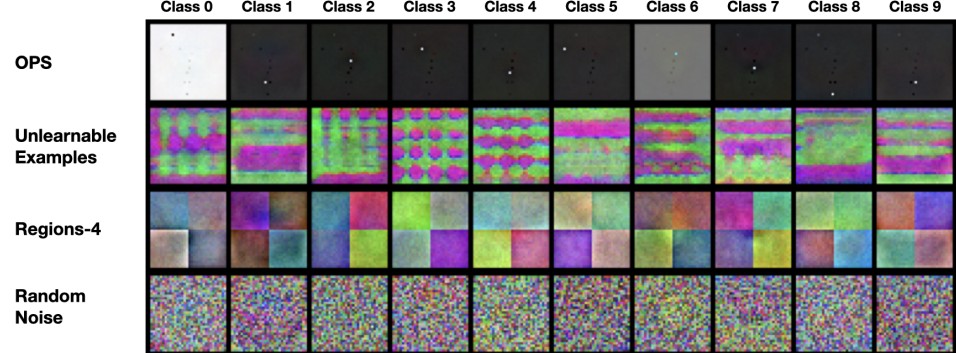

Figure 7: **Learned Weights from first step of Orthogonal Projection on class-wise perturbed CIFAR-10 unlearnable datasets.** We visualize learned weights ($W$ in Algorithm 1) after training a linear model on unlearnable datasets. Learned weights appear to recover the added class-wise perturbation for all datasets. In our Orthogonal Projection attack, we project each perturbed image to be orthogonal to each of these learned weights (Algorithm 1, Line 6).

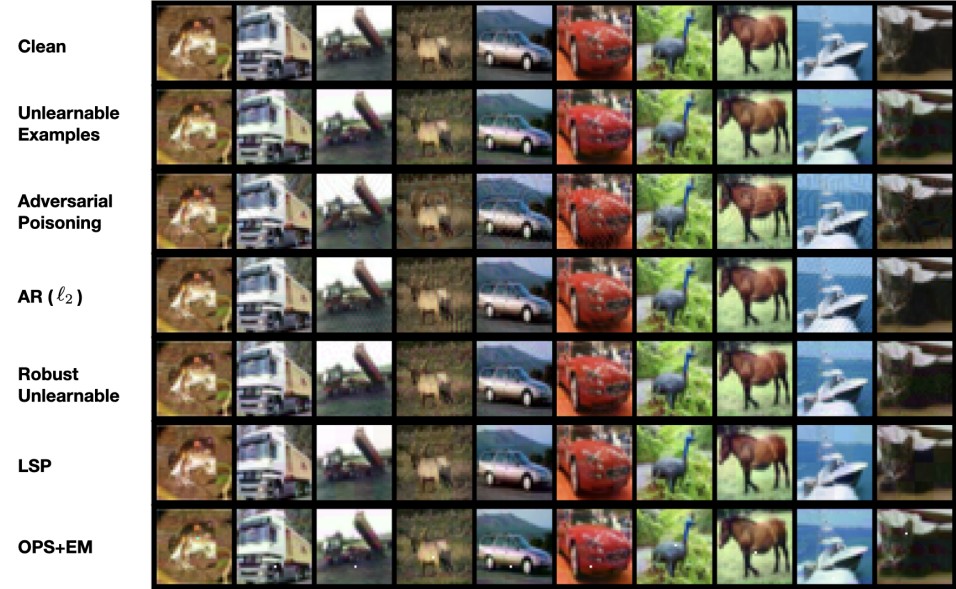

Figure 8: **Samples from CIFAR-10 unlearnable datasets.** We visualize the first 10 images from each sample-wise perturbed unlearnable dataset.

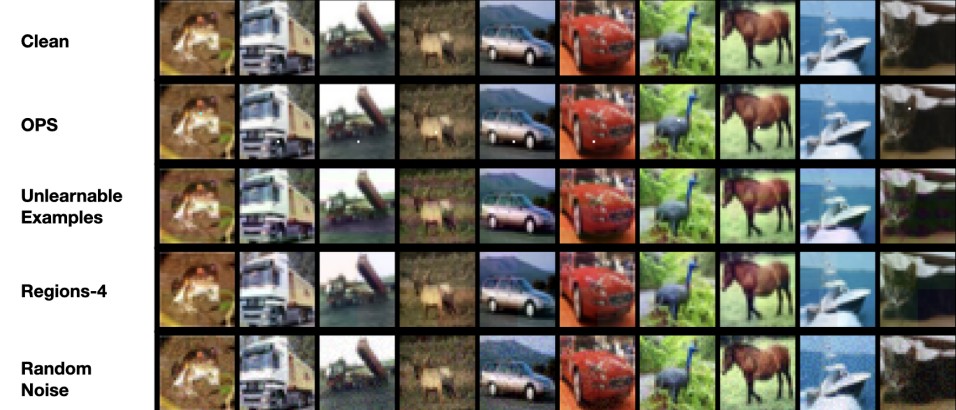

Figure 9: **Samples from CIFAR-10 unlearnable datasets.** We visualize the first 10 images from each class-wise perturbed unlearnable dataset.

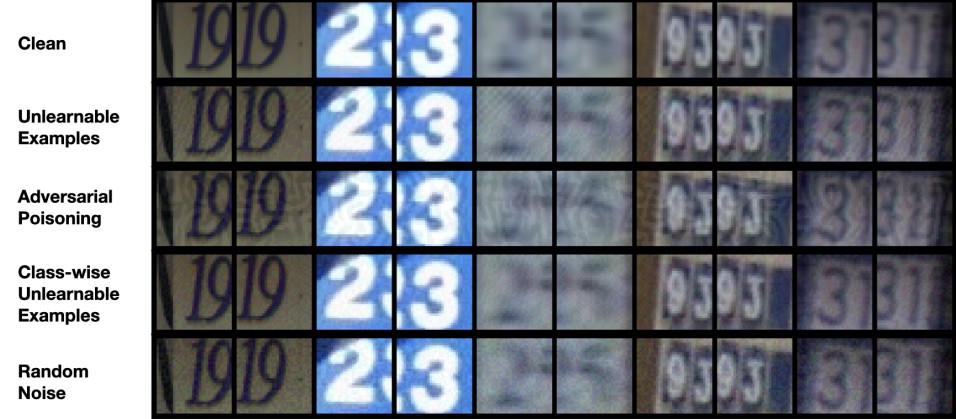

Figure 10: **Samples from SVHN unlearnable datasets.**

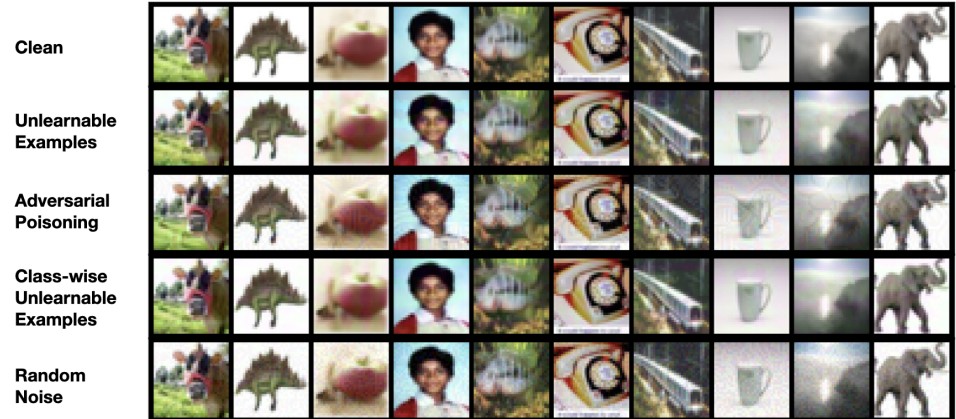

Figure 11: **Samples from CIFAR-100 unlearnable datasets.**

