# OpenReview forum: "What Can We Learn from Unlearnable Datasets?"
_NeurIPS.cc/2023/Conference — NeurIPS 2023 poster_

### Official Review · Reviewer_zKV8 · 2023-06-26

**Soundness:** 4 excellent
**Presentation:** 4 excellent
**Contribution:** 4 excellent
**Rating:** 7
**Confidence:** 5

**Summary:**

This paper comprehensively evaluated existing unlearnable examples and showed a surprising result. Unlearnable examples aim to prevent the model from learning useful features from the data. However, results show that several methods that networks actually can learn useful features. This is revealed by applying an existing feature reweighting method. Such results indicate that "privacy" might not be preserved. This paper also demonstrated that the recent findings of linear separation in unlearnable examples are not a necessary condition and an orthogonal projection attack can break class-wise linear separable unlearnable noise.

**Strengths:**

- By using an existing method of Deep Feature Reweighting (DFR), the finding is very interesting and new in this field. This paper has comprehensively evaluated the existing different methods for generating unlearnable examples. It is surprising to see model can "actually" learn useful features on several unlearnable methods. Interestingly, the error-minimizing does not. Although the exact cause remains unknown, such findings provide valuable insights for future works. Additionally, this also indicates that future works should consider DFR as a reasonable evaluation method for unlearnable examples.
- Based on recent work, Yu et al. [34] show that unlearnable examples rely on linearly separable noise. This paper extended the analysis to a wide range of different generation methods. Results show that although unlearnable examples are commonly induced linear separable noise, one particular method, Autoregressive perturbations, does not fall into this category. This indicates that linear separable noise is not a necessary condition. These results provide insights for future works that there could be more effective non-linear separable unlearnable noise.
- The presentations in this work are very good. It covered comprehensive related works, motivations, challenges and limitations in unlearnable examples.

**Weaknesses:**

My main concern is the Orthogonal Projection attack. Although it is technically sound, it is unclear what is the main contribution or the insight of this proposed method.
- Class-wise noise is known to be easily detected in unlearnable examples. Considering an additive noise, Yu et al. [34] show averaging across classes can easily expose the class-wise noise. One trivial solution that might consider is to subtract the noise for each class. Does an Orthogonal Projection attack more effective compared to this trivial solution? For Orthogonal Projection attacks, does it have any effect on the sample-wise setting?
- Compared to adversarial training (AT), the main benefit is efficiency. However, AT has other desired properties, such as adversarial robustness and learning robust features. As for effectiveness, it seems to be no significant benefit of the Orthogonal Projection attack. For OPS or OPS+EM, the constraint is a single pixel, while AT evaluated in this paper is $L_\infty$, changing AT constraint might be effective against the OPS-based method.

**Questions:**

See the weaknesses section.

**Limitations:**

The authors adequately addressed the limitations and potential negative societal impact.

---

> ### Author Rebuttal · Authors · 2023-08-10
>
> We thank the reviewer for their thorough review, for mentioning that our findings are “very interesting and new in this field,” and for writing that our paper provides “valuable insights for future works.”
>
> > Class-wise noise is known to be easily detected…Yu et al. [34] show averaging across classes can easily expose the class-wise noise. One trivial solution that might consider is to subtract the noise for each class. Does an Orthogonal Projection attack more effective compared to this trivial solution?
>
> We did perform experiments averaging images of a given class to recover class-wise perturbations, but simply subtracting this class image from each image doesn’t remove the poisoning effect. Additionally, because we do not know the true class at inference time, we cannot subtract the class image (and so there is also a distribution mismatch between train/test). This trivial method of subtracting average class images is compared to Orthogonal Projection below:
>
> | Poison | Class-Avg Subtract | Orthogonal Projection |
> |--------|--------------------|-----------|
> | LSP    | 13.05              | 87.99     |
> | OPS    | 12.62              | 87.94     |
>
> Here, we report CIFAR-10 test accuracy for LSP and OPS poisons. We will be sure to add these results to the Appendix for completeness. Also, we revisited Yu et al. [34] but did not find their experiment where they average images of a class. Please let us know if there is a different work you were referring to, and we would be happy to comment.
>
> > For Orthogonal Projection attacks, does it have any effect on the sample-wise setting?
>
> Orthogonal Projection sometimes works on sample-wise perturbations (for example, NTGA and Unlearnable Examples in Table 3), but is not meant for this case since perturbations can vary greatly within a class. This can make it difficult to optimize a representative set of vectors (columns of W in Alg. 1 Line 3) which are predictive of the label.
>
> > Compared to adversarial training (AT), the main benefit is efficiency…AT has other desired properties, such as adversarial robustness and learning robust features.
>
> We agree there are desirable properties of adversarial training (AT). For learning from unlearnable data, however, the goal is only to improve clean test performance – to somehow learn from corrupted data and generalize to clean data. Aside from being ~5x faster than AT,   Orthogonal Projection is also more interpretable: we can visualize the most predictive features (Figure 3 (b)) and choose whether to remove these features. On the other hand, it is difficult to understand which features AT is ignoring.
>
> Additionally, the OPS poison is a case where AT will fail due to the unbounded perturbation, but Orthogonal Projection can achieve 87.94% test accuracy, which is higher than 14.41% of AT, and higher than 85.16% of ISS [37].
>
> [37] Image Shortcut Squeezing: Countering Perturbative Availability Poisons with Compression, ICML 2023
>
> > changing AT constraint might be effective against the OPS-based method
>
> The authors of OPS [33] indeed conduct an experiment where they perform $L_2$ AT as opposed to $L_{\infty}$ and find that the poisoning effect can be mitigated: Using $\epsilon=2$ $L_2$ AT yields a model with 73.7% test accuracy. But this is at the cost of clean test accuracy which degrades from 94% to 73.45%. Orthogonal Projection is more effective as it does not degrade clean accuracy much (90% test accuracy on clean train data), and achieves high test accuracy (87.94%) on OPS poison.

---

> > ### Comment · Reviewer_zKV8 · 2023-08-13
> > **Thanks for the clarification**
> >
> > Thanks for the clarification. My concerns have been addressed. I very much appreciate this paper.

---

### Official Review · Reviewer_vqjk · 2023-07-04

**Soundness:** 3 good
**Presentation:** 2 fair
**Contribution:** 2 fair
**Rating:** 4
**Confidence:** 4

**Summary:**

This paper suggests that DNNs can learn useful features from unlearnable datasets and provides a counterexample, demonstrating that linear separability of perturbations is not a necessary condition. They propose the Orthogonal Projection method to recover unlearnable datasets.

**Strengths:**

1. A new method called Orthogonal Projection is proposed to recover unlearnable datasets. On class-wise unlearnable datasets, this method achieves similar (or even betters) performance compared to adversarial training. Moreover, it is much faster than adversarial training.
2. This work suggests the risk of using class-wise perturbations to craft unlearnable datasets.

**Weaknesses:**

1. For the claim "DNNs can learn useful features from unlearnable datasets", your experimental evidence is not sufficient and the logic is not very solid. See Question 1.
2. Though the intuition of Orthogonal Projection is clear, the purpose of its mathematical form is still ambiguous. Especially it cannot explain why this method is good at treating class-wise perturbations rather than sample-wise ones. See Question 2.
3. Sections 4.2, 4.3 and 4.4 have weak connections with each other, appearing as if they are independent from one another. The results in sections 4.2 and 4.3 are not interesting enough.

**Questions:**

1. About weakness 1:
1.1 In Figure 2, the performance of DFR on different unlearnable datasets is not consistent. At the last epoch, DFR in (b, d) is even worse than randomly initialized baseline. For certain datsets, such as Random noise and AR, the peak in early epochs is a property of these datsets, not your DFR method.
1.2 I don't think the increase of test accuracy caused by DFR can support you claim because you retrain the linear classifier using clean data. Don't forget that your DFR method improve the accuracy of randomly initialized baseline from $10\%$ to $35.79\%$. However, you can not conclude that a randomly initialized feature extractor learns useful features.
2. About weakness 2:
2.1 In the algorithm of Orthogonal Projection, the recovered data $X_r$ is ensured that $W^T X_r=0$. What does this equation mean? How does it work for recovery?
2.2 What is the reason that Orthogonal Projection is not good at recovering sample-wise perturbations?
2.3 It is mentioned in Line 262 that perturbed images of sample-wise Unlearnable Examples and Adversarial Poisoning are not linearly separable though perturbation themselves are linearly separable. Does this illustration contradict to the claim in Line 216 "learning the most predictive features in an unlearnable dataset can amount to learning the perturbation itself"?
2.4 Please give experimental evidence of linear (in)separability of images perturbed by those class-wise and sample-wise attacks.


**Limitations:**

As mentioned in the paper, orthogonally projection removes K dimensions from the data manifold, where K is the number of classes in a dataset. While this may not be a problem for high-resolution images with tens of thousands of dimensions, this detail could impact applicability for low-resolution datasets.
Also see the weakness.

---

> ### Author Rebuttal · Authors · 2023-08-10
>
> We thank the reviewer for their thorough feedback and for recognizing that our work “suggests the risk of using class-wise perturbations.” That was one of our goals in this work.
>
> > About weakness 1: 1.1 In Figure 2, the performance of DFR on different unlearnable datasets is not consistent. At the last epoch, DFR in (b, d) is even worse than randomly initialized baseline. For certain datasets, such as Random noise and AR, the peak in early epochs is a property of these datasets, not your DFR method.
>
>
> This behavior is due to the different poison generation methods. More specifically, adversarial poisoning perturbs images with error-maximizing noise (and this noise has been shown to inject actual features models use during classification [36]). In this case, DFR reweights useful existing features for classification. On the other hand, Random Noise and AR poisons do not perturb images with image features; instead, both perturb with synthetic noise. In these cases, useful features are still learned during poison training, but only in the first epochs of training. In Figure 2, plots (a) and (d), we find that checkpoint features progressively get better despite test accuracy of the model remaining low. In plots (b) and (c), we find that checkpoint features are better around the test accuracy peak. In both cases, features learned during poison training can be reweighted for high test accuracy – calling into question the ability for any of these unlearnable dataset methods to safeguard data.
>
> > 2.1…What does this equation mean? How does it work for recovery? 2.2 What is the reason that Orthogonal Projection is not good at recovering sample-wise perturbations?
>
> Assume CIFAR-10 images of shape (3,32,32). Each column i of W (optimized in Alg. 1, Lines 1-4) is a 3072-dim vector that represents the most predictive image feature for class i. This step serves as recovery of the perturbation. After the QR decomposition of W, Q consists of orthonormal columns that form a basis for the column space of W. When we say Orthogonal Projection “ensures that the dot product of a row of X with every column of Q is zero,” (i.e., $X_r \cdot Q = 0$) this means that every recovered image vector does not contain any linearly separable component (i.e., does not contain any column of Q as a component). Alg. 1, Line 6 ensures image vectors and columns of Q are orthogonal and so the dot product is 0. The “recovered” data thus has 10 dimensions (approximations of the 10 perturbations) removed. We have now updated Sec. 4.1.1 to be more clear on these steps, and we will include these updates in our camera ready version.
>
> Orthogonal Projection sometimes works on sample-wise perturbations, but is not meant for this case since perturbations can vary greatly within a class. This can make it difficult to optimize a representative set of vectors (W) which are predictive of the label.
>
> > 2.3 It is mentioned in Line 262 that perturbed images of sample-wise Unlearnable Examples and Adversarial Poisoning are not linearly separable though perturbation themselves are linearly separable. Does this illustration contradict to the claim in Line 216 "learning the most predictive features in an unlearnable dataset can amount to learning the perturbation itself"?
>
> We don’t claim that learning the most predictive features will “amount to learning the perturbation itself”, only that it can amount to that. More specifically, the many poisons we consider in our work all behave differently, having been generated or optimized with a variety of objectives. Despite this, we find that for class-wise noises, a logistic regression classifier can learn the most predictive features and thereby isolate the perturbation itself (Figure 3).
>
> > 2.4 Please give experimental evidence of linear (in)separability of images perturbed by those class-wise and sample-wise attacks
>
> In the paper, we document the linear separability of **perturbations** from various poisons, as in [34]. Perturbed images, on the other hand, behave differently. Prompted by your feedback, we report logistic regression train accuracy on various CIFAR-10 poison images below, and will include these results in our camera ready version:
>
> |   Data                    |   Train Accuracy |
> |---------------------------|------------------|
> |   Clean                   |   53.88          |
> |   Unlearnable Examples    |    100        |
> |   Adversarial Poisoning   |     49.47       |
> |   AR (L2)                 |      37.99     |
> |   Robust Unlearnable      |       67.21     |
> | LSP                       | 100            |
> | OPS+EM                    | 100            |
> | (CW) OPS                  | 100          |
> | (CW) Unlearnable Examples | 100           |
> |   (CW) Random Noise       | 100          |
>
> In the table above, we find that Unlearnable Examples, LSP, OPS, and class-wise (CW) poisons have linearly separable poison images, but the remaining poisons we consider do not.
>
> Thank you again for your thoughtful review. We made a significant effort to address your feedback including experiments and multiple paper edits, and we would appreciate it if you would consider raising your score in light of our response. Do you have any additional questions we can address?

---

> > ### Comment · Reviewer_vqjk · 2023-08-17
> >
> > I appreciate the efforts made by the authors to answer my questions.
> >
> > Firstly, your clarification helped me understand why your proposed orthogonal projection method is effective, especially in defending against class-wise poisoning. Taking CIFAR-10 as an example, we can approximate that the 10 column vectors of Q represent 10 class poisons. Your method precisely makes the processed images orthogonal to that 10-dimensional subspace. I find this idea reasonable.
> >
> > However, this also highlights the limitation of the method, which is the inability to guarantee the effectiveness for sample-wise poisoning, especially considering some recently proposed defense methods [a, b] that achieve better defense results. Perhaps focusing only on defense against class-wise poisoning would be helpful.
> >
> > Secondly, I still believe that Sections 4.2 and 4.3 are not sufficiently interesting and have weak connections to the most important Section 4.4 in the overall article.
> >
> > Therefore, I decided to keep the rating.
> >
> > [a] Image Shortcut Squeezing: Countering Perturbative Availability Poisons with Compression. ICML 2023
> >
> > [b] Learning the Unlearnable: Adversarial Augmentations Suppress Unlearnable Example Attacks. arXiv 2023.

---

> > > ### Author Response · Authors · 2023-08-19
> > > **Clarifications on importance of findings in response to comment**
> > >
> > > > limitation of the method, which is the inability to guarantee the effectiveness for sample-wise poisoning, especially considering some recently proposed defense methods [a, b]
> > >
> > > Given that class-wise poisons are still being developed in numerous works [10, 33, 34, 18], we believe research would benefit from our novel benchmark attack (Orthogonal Projection). We believe that without knowledge of our results, class-wise poisons could continue to be developed without a strong baseline to compare against. In fact, one of the reasons we developed the Orthogonal Projection method was because we could not find a good explanation for why papers opted for sample-wise poisons when class-wise poisons were more effective [10]. Our work demonstrates that removing a class-wise perturbation is not as simple as taking the average class image (see rebuttal pdf), and that perturbations can be recovered, as we illustrate in Figure 3.
> > >
> > > While ISS [b] is effective, there is little theoretical evidence to suggest that this effectiveness is “guaranteed.” While the authors explain that “grayscale compression is used to eliminate low-frequency shortcuts, and JPEG compression is used to eliminate high-frequency shortcuts,” a DCT transform is never performed to analyze whether this is occurring or not. On the other hand, vector projections and orthogonal subspaces can be reasoned about for images theoretically, and tested empirically as we do in Section 4.4 and Appendix A.3. Compared to ISS [b], our Orthogonal Projection method produces higher test accuracy on the recently published class-wise OPS poison (87.94% (ours) vs 85.16% [b]).
> > >
> > > [10] Unlearnable Examples: Making Personal Data Unexploitable, ICLR 2021
> > >
> > > [18] Learnability lock: Authorized learnability control through adversarial invertible transformations, ICLR 2022
> > >
> > > [...] remaining citations are from paper references
> > >
> > > > Sections 4.2 and 4.3 are not sufficiently interesting
> > >
> > > Section 4.2 presents a new way of analyzing learned representations of poisoned models. Deep feature reweighting allows us to probe poisoned models for useful features and evaluate unlearnable dataset methods in a new way. We find it interesting that some unlearnable datasets produce checkpoints that progressively improve test accuracy during poison training.
> > >
> > > Section 4.3 presents a counterexample to the linear separability hypothesis [34]. Without knowledge of this result, one could be inclined to believe that data poisoning is extremely simple, relying on only linearly separable perturbations despite a wide variety of different optimization objectives and theory used to generate/optimize the poisons. We find it interesting that newer methods [26] can be more complex (not linearly separable) and still work as poisons. This result could inspire future research on non linearly separable perturbations, given that necessary conditions for unlearnable datasets remains an open problem.
> > >
> > > > weak connections to the most important Section 4.4
> > >
> > > We can see how the paper’s organization could suggest a deeper connection between our findings in 4.2 and 4.3 and the Orthogonal Projection method. But, as we state in the abstract and introduction, the goal of our paper is “make a number of findings that call into question [Unlearnable Datasets’] ability to safeguard data.” Each section of the paper (4.2 to 4.4) is a finding that, we believe, can be viewed on its own and can inform the poisoning community about a fundamental issue of current unlearnable dataset methods.
> > >
> > > For example, Unlearnable Datasets can be seen as not suitable to protect data because one can learn generalizable features from poisoned data (Section 4.2). On the other hand, Unlearnable Datasets can be seen as not suitable to protect data because class-wise perturbations are still commonly used and we can develop a method to find and remove them (Section 4.4).
> > >
> > > —
> > >
> > > Thank you again for taking the time to read and discuss our work with us. We appreciate your insights and have made additional paper edits following this response, in addition to the new table from your previous suggestion in this thread. Are there other specific suggestions that, if addressed, would potentially warrant a reconsideration of the current rating? Your guidance would greatly assist us in refining the paper's content.

---

### Official Review · Reviewer_kEiu · 2023-07-05

**Soundness:** 3 good
**Presentation:** 4 excellent
**Contribution:** 3 good
**Rating:** 4
**Confidence:** 4

**Summary:**

This paper conducts an analysis of the properties of unlearnable dataset methods to evaluate their potential for future viability and security assurances. It is demonstrated that neural networks possess the ability to learn generalizable features from unlearnable datasets, while also suggesting that image privacy may not be effectively preserved. Additionally, a counterexample is provided to challenge the widely held belief that unlearnable datasets induce learning shortcuts through the linear separability of added perturbations. To address this issue, an orthogonal projection attack is proposed, which enables learning from various unlearnable datasets. The results of this approach demonstrate that linearly separable perturbations should not be relied upon.

**Strengths:**

1. The originality of the paper is good, as it gives us a different view to unlearnable examples and poisoning attacks by
    (a). Demonstrating that neural networks possess the ability to learn generalizable features from unlearnable datasets
    (b). Giving a counterexample is provided to challenge the widely held belief that unlearnable datasets induce learning shortcuts through
      the linear separability of added perturbations
    (c). Proposing a new evaluation framework and a novel attack method to assess the viability and security promises of unlearnable datasets.
   Overall, The paper's findings challenge some widely held beliefs about unlearnable datasets and provide insights into their limitations and
   potential vulnerabilities
2. The proposed method can achieve good results on some of unlearnable tasks (class-wise) and is more effective than adversial training.
3. This paper is well-writen

**Weaknesses:**

1. The motivation for the paper does not align to me. I hope that the proposed method is motivated by the findings (Sections 4.2 and 4.3), however, the authors only illustrate the relationship between linearly separable perturbations and the proposed orthogonal projection method, making Section 4.2 a superfluous. There should be more discussion on the relationship between the findings and the methods presented in Section 4.4.
2. The author uses the DFR method to prove that DNN can learn useful features, but it is not convincing to me. The improvement in accuracy may simply come from using cleaner samples. (eg Figure 2, pictures (b), (c)). Also, I think the act of loss should also be involved in this part. There should be more references to why "the higher the test accuracy after DFR, the more likely the model will pick up on private image features present in the original clean data".
3. Although the authors give a counterexample to challenge the commonly held view that non-learnable datasets induce learning shortcuts by adding perturbed linear separability, it lacks theoretical analysis.
4. The method only achieves satisfactory results on Class-Wise perturbation, but the generalization of the method is not enough. In contrast, adversarial training has better generalization ability.
5. Limited evaluation: Evaluation requires more datasets.



**Questions:**

1. The correlation between the method proposed in Section 4.4 and the findings in Sections 4.3 and 4.2, particularly 4.2, is of interest.
2. Can additional sources be provided to corroborate the assertion that "the higher the test accuracy after DFR, the more likely it is that the model has earned private image features present in the original clean data"?
3. The efficacy of the proposed method on other datasets merits investigation. It is recommended that further experimentation be conducted.
4. The dissimilar trends displayed by (a) (d) and (b) (c) in Figure 2 require explanation.

**Limitations:**

Yes, the authors adequately addressed the limitations.
However, in order to further enhance and augment the credibility and validity of their work, it is suggested that they provide a more elaborate and comprehensive explanation regarding the various methodologies and motivations that were employed in their research process. Additionally, it would be highly beneficial if they could include a greater number of experiments and references within their work, as this would serve to substantially bolster and fortify the robustness and soundness of their findings and conclusions.

---

> ### Author Rebuttal · Authors · 2023-08-10
>
> We thank the reviewer for their careful review, for mentioning that our work “gives us a different view to unlearnable examples,” and for recognizing that our results “challenge some widely held beliefs about unlearnable datasets.” Those were our goals in this work.
>
> > The author uses the DFR method to prove that DNN can learn useful features, but it is not convincing to me. The improvement in accuracy may simply come from using cleaner samples
>
> By finetuning a randomly initialized network (first row of Table 1) on a random subset of 5k clean samples, we can measure the effect of the clean samples. By doing the same procedure on poisoned networks and measuring the improvement **relative to** the finetuned randomly initialized network, we can measure the utility of poisoned-network features. Looking at Figure 2, plot (b), suppose we obtain checkpoints by training on class-wise random noise at epochs 3 and epoch 40. If we finetune the epoch 3 checkpoint (with 5k clean samples) we get test accuracy of 66.8%, while if we finetune epoch 40 checkpoint (with the same 5k clean samples) we get test accuracy of 19.2%. Relative to a finetuned random-init checkpoint (35.9% test acc), epoch 3 features obtained by training on the perturbed unlearnable data, yield more than a 30% increase in test accuracy. Useful features, which help test set generalization, are therefore present at epoch 3 but not at epoch 40. We found it surprising that despite both checkpoints being trained on unlearnable data, the learned features could still rise to such high performance.
>
> > The method only achieves satisfactory results on Class-Wise perturbation, but the generalization of the method is not enough
>
> Our Orthogonal Projection method is designed to exploit class-wise perturbations by design. It is not meant to be a general solution. Nearly all Unlearnable Dataset papers focus on optimizing sample-wise noise [10,7,26,35,27,34,33], as opposed to class-wise noise, with little explanation why. We designed Orthogonal Projection to answer, for ourselves, why class-wise perturbations should not be relied on. In [10], the authors note that despite class-wise error-minimizing noise being “superior to random noise” and “more efficiently and more flexibly in practical usage,” the noise “may get more easily exposed.” In [26], the authors state that “because class-wise perturbations can be recovered by taking the average image of a class, these should therefore be easy to remove” but do not provide evidence. If class-wise noises are more easily exposed, how does one remove them? We found that although one can average a class image, it is difficult to train a new network to reasonably high test accuracy by removing the average image. Instead, our solution is to project image features orthogonal to features optimized by logistic regression. Our Orthogonal Projection method is specifically for class-wise perturbations, which have recently been used to lock and protect data [18].
>
> [10] Unlearnable Examples: Making Personal Data Unexploitable, ICLR 2021
> [18] Learnability lock: Authorized learnability control through adversarial invertible transformations, ICLR 2022
> [26] Autoregressive Perturbations for Data Poisoning, NeurIPS 2022
> [...] remaining citations are from paper references
>
> > "the higher the test accuracy after DFR, the more likely it is that the model has learned private image features present in the original clean data"
>
> Unlearnable dataset perturbations are supposed to stop the model from learning useful features, and if the model gets high test accuracy by training on them, that demonstrates that it has learned useful features. Thank you for pointing out our ambiguous language.  We have updated our draft to clarify and to remove the confusing “private image features” terminology.  We will include this update in our camera ready version.
>
> > The efficacy of the proposed method on other datasets merits investigation. It is recommended that further experimentation be conducted.
>
> In the Supplementary Material, we include additional experiments on SVHN, CIFAR-100, and an ImageNet subset (Appendix 2.2) to enhance the results of Section 4.2. Moreover, we include additional experiments on CIFAR-100 in Appendix 3.2 to augment Section 4.4. We also provide additional results for different model architectures in Appendix 2.1 and Appendix 3.1.
>
> >The dissimilar trends displayed by (a) (d) and (b) (c) in Figure 2 require explanation.
>
> Thank you for pointing this out. We have added additional explanation to the text following Figure 2. To clarify: This behavior is due to the different poison generation methods. More specifically, adversarial poisoning perturbs images with error-maximizing noise (and this noise has been shown to be actual features models use during classification [36]). In this case, DFR is reweighting useful, existing features for classification, leading to higher test accuracy. On the other hand, Random Noise and AR poisons do not perturb images with useful features; instead, both perturb with synthetic noise. In these cases, useful features are still learned during poison training, but only in the first epochs of training. As training progresses, the model checkpoints are continually corrupted by synthetic noise features which cannot be useful for classification despite reweighting. By “useful,” we mean the technical definition in Eq. 1 [36].
>
> [36] Adversarial Examples Are Not Bugs, They Are Features, NeurIPS 2019
>
> Thank you again for your thoughtful review. We made a significant effort to address your feedback including multiple paper edits, and we would appreciate it if you would consider raising your score in light of our response. Do you have any additional questions we can address?

---

> > ### Comment · Reviewer_kEiu · 2023-08-16
> >
> > While your previous explanation helped address some of my questions, there are still a couple major issues that need to be better clarified:
> >
> > 1) It would be helpful to see a stronger link between the orthogonal projection method proposed in Section 4.4 and the results from Sections 4.2 and 4.3, especially 4.2. The proposal seems like it should be motivated by those earlier findings, but right now the paper only shows the connection to linearly separable perturbations, making Section 4.2 seem unnecessary. Please illustrate more clearly how the observations in 4.2 and 4.3 led to the proposed technique.
> >
> > 2) For the weakness 2, the evaluation should include more experiments and metrics beyond just test accuracy, such as loss over training, Relying solely on test accuracy doesn't fully support the claim. Including additional graphs would provide a more thorough analysis.

---

> > > ### Author Response · Authors · 2023-08-19
> > > **Clarifications and new metrics in response to comment**
> > >
> > > > Please illustrate more clearly how the observations in 4.2 and 4.3 led to the proposed technique
> > >
> > > We can see how the paper’s organization could suggest a deeper connection between our findings in 4.2 and 4.3 and the Orthogonal Projection method. But, as we state in the abstract and introduction, the goal of our paper is “make a number of findings that call into question [Unlearnable Datasets’] ability to safeguard data.” Each section of the paper (4.2 to 4.4) is a finding that, we believe, can be viewed on its own and can inform the poisoning community about fundamental issues of current unlearnable dataset methods.
> > >
> > > For example, Unlearnable Datasets can be seen as not suitable to protect data because one can learn generalizable features from poisoned data (Section 4.2). On the other hand, Unlearnable Datasets can be seen as not suitable to protect data because class-wise perturbations are still commonly used and we can develop a method to find and remove them (Section 4.4). In both cases, model parameters are optimized using unlearnable data – in Sec 4.2 the optimization yields generalizable features, while in Sec 4.4 the optimization yields the class-wise perturbation.
> > >
> > > We have added additional clarifications to the introduction, we would be open to suggestions as to re-organization of the findings to minimize confusion.
> > >
> > > > the evaluation should include more experiments and metrics beyond just test accuracy, such as loss over training
> > >
> > > Following your advice, we have measured the train and test loss of each poison checkpoint after DFR and compared it to the loss of a randomly initialized model after DFR. As a reminder, both these losses are computed when evaluating poisoned checkpoints on clean train/test data. The following table performs the same experiment as in Table 1, Section 4.2, where we investigate what poisoned models learn. We use the same random subset of 5,000 clean CIFAR-10 train samples from the paper for DFR.
> > >
> > > |   Train Data                          |   Min DFR Train Loss  |   Min DFR Test Loss  |
> > > |---------------------------------------|-----------------------|----------------------|
> > > |   None                                |   2.625               |   2.379              |
> > > |   Unlearnable Examples [10]           |   1.777               |   1.798              |
> > > |   Adversarial Poisoning [7]           |   0.966               |   1.036              |
> > > |   AR [26]                             |   1.408               |   1.531              |
> > > |   NTGA                                |   1.342               |   1.391              |
> > > |   Robust Unlearnable [27]             |   1.789               |   1.790              |
> > > |   LSP [34]                            |   1.705               |   1.675              |
> > > |   OPS+EM [33]                         |   1.857               |   1.869              |
> > > |   (CW) OPS [33]                       |   1.669               |   1.697              |
> > > |   (CW) Unlearnable Examples [10]      |   2.043               |   2.037              |
> > > |   (CW) Regions-4 [25]                 |   1.669               |   1.706              |
> > > |   (CW) Random Noise                   |   1.170               |   1.352              |
> > >
> > > In the first row, we report the train and test loss of a randomly initialized RN-18 after DFR. In the subsequent rows, we report min train/test loss of poisoned checkpoints after DFR. We find that the trends when analyzing the loss follow the trends when analyzing test accuracy: Firstly, Adversarial Poisoning, AR, NTGA, and (CW) Random Noise achieve lowest losses – and accordingly those poisoned checkpoints also have the highest DFR test accuracy in Table 1. Secondly, and more interestingly, we find that **all** poisoned models have a lower loss than the finetuned (DFR) random initialized model. This reinforces our claim that models learned useful features from poisoned data. Interestingly, (CW) Unlearnable Examples come closest to the expected random chance loss of -log(1/10)=2.302. This suggests (CW) Unlearnable Examples are best at corrupting network weights from finetuning. Note that train loss is slightly lower than test loss in many cases because train loss is computed over the entire train set (which includes the subset of 5k samples we used for DFR).
> > >
> > > Thanks again for your suggestions which have made our work more comprehensive. We have produced two additional plots (DFR train loss vs epoch and DFR test loss vs epoch), and a new table which we will include in our camera-ready version. We would appreciate it if you would consider revising your score in light of our response. And do let us know if you have any additional questions.

---

### Official Review · Reviewer_TH5w · 2023-07-09

**Soundness:** 4 excellent
**Presentation:** 4 excellent
**Contribution:** 3 good
**Rating:** 7
**Confidence:** 5

**Summary:**

This paper studies the problem of the actual learnability of unlearnable datasets. Specifically, the authors have demonstrated that unlearnable datasets that are generated by existing methods can actually be used to learn generalizable features. In addition, the authors show that it is not necessary to make poisons linearly separable for achieving effective poisoning effects. Furthermore, based on the fact that most existing poisoning methods rely on linear separability, the authors propose a simple yet effective countermeasure to recover clean images for training.

**Strengths:**

- It is valuable to revisit existing work in a popular field like data poisoning.
- The paper is very well written, with a thorough review of related work and sufficient descriptions of technical details.
- Experiments are extensive in terms of the number of models, datasets, settings, compared baselines, and so on.
- Several visualizations are provided to help support new findings.

**Weaknesses:**

I very much appreciate the paper, and there are no major weaknesses but minor ones about tuning down some claims:

1. This paper is not the first to find a counterexample to the linear separability hypothesis considering that previous work has intentionally relied on ideas beyond linear separability for generating unlearnable datasets [a].

2. The orthogonal projection method is not the only simple yet effective countermeasure considering [b], where the simple image compression-based method, ISS, is applied to remove poisoning perturbations. About the results, it seems that ISS is more globally effective than the orthogonal projection method. In addition, It would also be appreciated if diffusion models [3] and error-maximizing augmentation [20] can be compared. These two methods are also conceptually simple because they rely on either a per-trained (diffusion) model for pre-processing or data augmentation (that may not necessarily be adversarial).

[a] Is Adversarial Training Really a Silver Bullet for Mitigating Data Poisoning? ICLR 2023

[b] Image Shortcut Squeezing: Countering Perturbative Availability Poisons with Compression. ICML 2023

**Questions:**

N/A

**Limitations:**

Yes.

---

> ### Author Rebuttal · Authors · 2023-08-09
>
> We thank the reviewer for their time and feedback. We appreciate your mentioning that our “experiments are extensive'' and that the paper "is very well written.”
>
> > previous work has intentionally relied on ideas beyond linear separability for generating unlearnable datasets [a]
>
> Thank you for letting us know about the work of [a], which is relevant to our findings in Section 4.2. While the authors of [a] motivate their approach using separability of recent poisoning perturbations, they only use t-SNE visualizations, which is different from the separability experiment we perform.
>
> More specifically, t-SNE cluster separability should not be equated to the linear separability we measure in Table 2 because it is possible to have linearly separable data that, when plotted using t-SNE, appears not separable. In other words, the poison from [a] could still contain linearly separable perturbations.
>
> > ISS is more globally effective than the orthogonal projection method
>
> Our Orthogonal Projection method is designed to exploit class-wise perturbations by design. It is not meant to be a general solution like ISS [b]. Nearly all Unlearnable Dataset papers focus on optimizing sample-wise noise [10,7,26,35,27,34,33], as opposed to class-wise noise, with little explanation why. We designed Orthogonal Projection to answer, for ourselves, why class-wise perturbations should not be relied on. Compared to ISS [b], our Orthogonal Projection method produces higher test accuracy on the recently published class-wise OPS poison (87.94% (ours) vs 85.16% [b]).
>
> [...] remaining citations are from paper references

---

> > ### Comment · Reviewer_TH5w · 2023-08-14
> > **Thanks for the rebuttal!**
> >
> > My concerns are well addressed. The authors are encouraged to incorporate the above discussions into the final version.

---

### Author Rebuttal · Authors · 2023-08-10

We'd like to thank everyone again for their reviews. A few reviewers mentioned wanting to see average images of a class to compare them to the learned weights from the first step of Orthogonal Projection. In the attached PDF, we include an additional figure which performs this visualization for two class-wise poisons. We will include this update in our camera ready version.

---

### Decision · Program_Chairs · 2023-09-21

**Decision:**

Accept (poster)

**Comment:**

This paper looks at different proposals for making unlearnable datasets, and shows that they fail (i.e., it remains possible to learn generalizable features from these datasets).
The reviewers had mixed opinions on the generalizability and novelty of the paper's contributions.
I encourage the authors to follow the reviewers' recommendations to more clearly compare their approach's strengths and weaknesses with those of adversarial training.